# Combating Human Viral Diseases: Will Plant-Based Vaccines Be the Answer?

**DOI:** 10.3390/vaccines9070761

**Published:** 2021-07-08

**Authors:** Srividhya Venkataraman, Kathleen Hefferon, Abdullah Makhzoum, Mounir Abouhaidar

**Affiliations:** 1Virology Laboratory, Department of Cell & Systems Biology, University of Toronto, Toronto, ON M5S 3B2, Canada; kathleen.hefferon@alumni.utoronto.ca (K.H.); mounir.abouhaidar@utoronto.ca (M.A.); 2Department of Biological Sciences & Biotechnology, Botswana International University of Science & Technology, Palapye, Botswana; makhzouma@biust.ac.bw

**Keywords:** vaccines, VLPs, genetic engineering, epitopes, transgenic, agroinfiltration

## Abstract

Molecular pharming or the technology of application of plants and plant cell culture to manufacture high-value recombinant proteins has progressed a long way over the last three decades. Whether generated in transgenic plants by stable expression or in plant virus-based transient expression systems, biopharmaceuticals have been produced to combat several human viral diseases that have impacted the world in pandemic proportions. Plants have been variously employed in expressing a host of viral antigens as well as monoclonal antibodies. Many of these biopharmaceuticals have shown great promise in animal models and several of them have performed successfully in clinical trials. The current review elaborates the strategies and successes achieved in generating plant-derived vaccines to target several virus-induced health concerns including highly communicable infectious viral diseases. Importantly, plant-made biopharmaceuticals against hepatitis B virus (HBV), hepatitis C virus (HCV), the cancer-causing virus human papillomavirus (HPV), human immunodeficiency virus (HIV), influenza virus, zika virus, and the emerging respiratory virus, severe acute respiratory syndrome coronavirus-2 (SARS-CoV-2) have been discussed. The use of plant virus-derived nanoparticles (VNPs) and virus-like particles (VLPs) in generating plant-based vaccines are extensively addressed. The review closes with a critical look at the caveats of plant-based molecular pharming and future prospects towards further advancements in this technology. The use of biopharmed viral vaccines in human medicine and as part of emergency response vaccines and therapeutics in humans looks promising for the near future.

## 1. Introduction

Increasingly, plants are being used as vaccine biofactories for expressing antibodies as well as foreign antigens using genetic engineering technologies. Plants are inherently advantageous for the production of vaccines as they inexpensive to grow at a large scale in greenhouses or bioreactors. Orally delivered vaccines preclude the costs and the time involved in the purification of the antigens, by virtue of the plant biomass being directly consumed to confer immunity. Plants can express complex antigens while dispensing with the risk of carrying human pathogens or endotoxins inherent to the bacterial, insect, or mammalian cell systems [1]. Plant material can be easily freeze-dried and made into tablets at a low cost for oral consumption [2,3]. When ingested orally, the plant-derived vaccines are protected within the stomach by the plant cell wall while being released slowly in the gut. Therefore cold chain facilities do not need to stock and deliver the respective plant material and there is greater cost efficiency compared to conventional mammalian or fermentation-based vaccines.

There are three popular modalities of expressing heterologous protein molecules in plant cells, namely, stable nuclear expression, transient expression using non-replicative or viral vectors, and transplastomic expression within chloroplasts [4]. Of these, stable expression of transgenes represents the most conventional approach which involves the insertion of the foreign gene into the plant genome. However, this approach has inherent caveats such as position effects depending on the site of insertion, low level of expression of the foreign protein due to the low copy number of the transgene, protracted lengths of time required to generate stably transformed plant lines. On the other hand, nucleus-encoded antigens can be expressed transiently using vectors derived from plant viruses to enable rapid expression. Deconstructed viruses delivered into the plants by Agrobacterium containing T-DNA molecules harboring viral replicons can enable fast, highly scaled up expression of the foreign proteins at robust yields while complying with the existing GMP industrial practices. The most popular choice of viruses used to generate these vectors include the Tobacco Mosaic Virus (TMV), the Cowpea mosaic virus (CPMV), and the Potato virus X (PVX) which are RNA viruses whilst among DNA viruses, the most successfully used is that of the Bean yellow dwarf geminivirus [5]. In transplastomic expression, the transgene undergoes site-specific integration into the chloroplast genome (plastome) wherein there are no silencing molecular mechanisms or position effects [6] and the plastome has a high copy number leading to substantially increased yields over that of nuclear transformation. The plastome is inherited maternally [7] which precludes undesired flow of genes caused by pollination, thus ensuring biosafety. The transformation of chloroplasts has become a decisive technology in biopharming [8,9].

The foreign protein expressed as above could be specifically localized within cellular compartments such as chloroplasts, protein bodies, or endoplasmic reticulum (ER) [10]. Among these, the most appealing mode of expression is in the ER as it contains the glycosylation machinery required for post-translational modifications in addition to having a low content of proteases. Foreign protein localization within oil bodies or protein bodies can enable ease of purification while enhancing yields [11,12]. The most extensively used vaccine production strategies in plants involve (a) appropriate promoters like tissue-specific promoters, promoters induced by environmental parameters or high-strength constitutive promoters, (b) protein targeting to specific organelles, (c) codon optimization, (d) alternate polyadenylation signals, (e) use of leader sequences to enhance translational efficiency, and (f) a variety of expression vectors [13]. In contrast to DNA-based vaccines which pose the risk of insertion into the human genome and the potential for oncogenesis, plant-produced Virus-Like Particles (VLPs) displaying viral epitopes are far safer compared to attenuated viruses and animal virus vector-derived vaccines. Therefore, there is no threat of incomplete inactivation of attenuated virus preparations or undesirable host responses to animal virus vectors [14].

A recent comprehensive review of the use of plant-based vaccines for the prevention and cure of human viral diseases is lacking. The current review addresses the use of plant-based vaccines and therapeutic antibodies for prophylaxis and therapy of human viral diseases. In this context are discussed vaccines against viruses such as HBV, HCV, Influenza Virus, HPV, HIV, SARS-CoV-2, and Zika Virus. Specifically, the current global incidence of diseases caused by these viruses, their molecular features relevant to vaccine design, and the latest developments in the generation of plant-derived vaccines against these viruses are addressed. The review closes with a note on the caveats of plant-based vaccines and future perspectives. Through this article, we hope to bring to light the idea that plant-based vaccines show great promise in combating viral diseases and will be the major avenue for low-cost and easy administration of vaccines in developing countries in the future.

## 2. HBV Vaccines

Hepatitis B Virus (HBV) is the causative agent of severe disease burden globally in the human population. Currently, nearly 257 million people have been affected by HBV infection [15]. HBV causes serious liver disease including liver cirrhosis and hepatocellular carcinoma which results in 887,000 deaths every year [16]. HBV infection in its chronic state causes severe liver pathogenesis accompanied by a high degree of morbidity and mortality. Currently, the best treatment for HBV disease happens to be based on nucleoside analogs and interferon which achieve suppression of the virus precluding progression of the disease. Vaccination seems to be the most cost-viable approach to contain the spread of HBV infection as none of the available treatments cure HBV infection [17]. Several challenges exist with respect to HBV vaccination such as (1) the occurrence of vaccine non-responders at the rate of 5–10% of individuals, (2) the levels of antibodies generated in most people who have been vaccinated is not sufficiently protective, (3) the emergence and transmission of resistant HBV escape mutants induced by the vaccines themselves, and finally (4) the absence of efficacious therapy for chronic HBV patients. Therefore, new effective HBV vaccines are called for that would augment seroprotection among non- or low-responders and afford therapeutic immunization in chronic HBV patients.

The DNA genome of the HBV codes for 3 surface proteins namely, the S (small), M (medium), and L (large) proteins all generated from the same ORF [18]. The S sequence is shared by all of these surface polypeptides and the M protein possesses the preS2 domain at its N-terminus whereas the L protein overlaps the M protein in addition to being lengthened by the preS1 domain. All of the said proteins are incorporated within the viral envelope and they assemble into 20 nm subviral particles containing the S protein as its major constituent [19]. Heterologous systems expressing the S protein by themselves form the HBsAg (HBV surface antigen) particles on which are founded most of the commercially available HBV vaccines [20,21]. The S protein possesses a conserved sequence between amino acids 122 and 150 which forms the external antigenic loop named as the ‘a’ determinant which is the main B-cell epitope [22].

HBV vaccine has been one of the most successful, trailblazing vaccines generated in modern times. In 1984, the recombinant HBV S antigen (S-HBsAg) was successfully expressed as a VLP vaccine in yeast [23] which was viable both in terms of safety and efficacy. However, there still is a compelling need for low-cost, non-parenteral vaccines amenable to oral delivery and herein the production of HBV vaccines in recombinant plant systems becomes significant. HBsAg was expressed in maize [24] and delivered orally into mice which showed enhancements in serum IgA levels and mucosal IgA titers in feces compared to the parenterally administered commercial Recombivax. The edible vaccine showed the induction of long-term memory as assessed by prolonged serum IgA and fecal IgA, IgG, and mIU/mL over the course of a year whereas the Recombivax showed only sustained IgG levels in the serum and mIU/mL, demonstrating that the oral vaccine provides long-term immunity both at the mucosal level and systemically. Therefore, the oral vaccine could act as an added layer of protection for sexually transmitted infections such as HBV which are contracted through mucosal surfaces. Rosales-Mendoza et al., 2015 [25], reported the expression of preS2-S antigen in carrots using signals for translocation and retention into the endoplasmic reticulum which showed enhanced levels of high expresser lines when compared to the preS2-S localized in the cytosol.

Deconstructed TMV-based cDNA MagniCON vectors (Icon Genetics, Halle, Germany) have been demonstrated to produce the greatest yield (300 mg/kg wet weight) of the small S HBsAg antigen in *Nicotiana benthamiana*. This venture was highly meritorious as the recombinant antigen exhibited the conformationally correct ‘a’ antigenic determinant and contained the full-length protein with disulfide-linked dimers that assembled into viable VLPs [26]. The Arizona Biodesign Institute was the first research group that established HBV vaccine biopharming [27]. This team showed that the middle M protein of the HBsAg was successfully generated in plants and injection of this antigen into mice stimulated a more robust B-cell immune response than the S protein [28]. They also expressed the HBcAg core antigen of HBV in MagniCON vectors wherein the antigen demonstrably assembled into highly immunogenic VLPs in plants and obtained a high yield of 2 g/kg wet weight [29].

The conventional HBV vaccine is based on the S protein and recent studies [17] have shown that the HBV-S/preS1^21−47^ antigen is a most favorable candidate and a good alternative for a future HBV vaccine. Dobrica et al., 2018 [30] reported *Lactuca sativa* (lettuce) plants transiently expressing the S/preS1^21–47^ antigen of HBV wherein oral administration of plant material without any adjuvant triggered antibody response in mice and these antibodies were capable of neutralizing HBV infection in HepG2-NTCP cell lines more efficiently than those elicited by mice that were fed on *Lactuca sativa* plants expressing the HBV S protein. This substantiates the use of the S/preS1^21–47^ antigen as a highly favored candidate for an edible HBV vaccine. This chimaeric protein contains the residues 21–47 of the L protein preS1 domain incorporated within the S protein external antigenic loop wherein the former plays an important role in the attachment of the virus to hepatocytes. This protein was generated successfully in *N. benthamiana* plants in which this novel antigen successfully assembled into subviral particles. These particles were shown to stimulate more robust humoral and cell-mediated immunity than the S protein and could be a better alternative to the poor response observed with the conventional HBV vaccine which is based on the S protein. A bivalent vaccine composed of a chimera of HBcAg VLPs displaying on their surface the Hepatitis E Virus ORF2 capsid protein immunological epitope was shown to be recognized by the anti-HBcAg mAb as well as the anti-HEV IgG positive swine serum when expressed in infiltrated *N. benthamiana* leaves using the pEAQ-HT vector [31].

## 3. HCV Vaccines

The world health organization (WHO) reports that about 130–150 million of the world’s population have chronic HCV infection with over 500,000 annual fatalities due to HCV-associated liver disease**.** A major proportion of infected individuals will develop chronic disease with a 15–30% risk of liver cirrhosis within 2 decades. At present, the only available therapeutic is antivirals notably, a combination of ribavirin and interferon which have been shown to be efficacious towards clearing all genotypes of HCV but are only effective in under 50% of the HCV patients [32]. Recently, other new drugs such as sofosbuvir [33] have become available which show great promise for circumventing HCV infection. The ‘‘Direct Acting Antivirals (DAAs)’’ [34] incur high costs and are available only in some developed nations. This necessitates the dire need for the generation of a safe and efficient HCV vaccine and anti-HCV vaccines remain the only viable options to prevent the occurrence of HCV infection. The main challenge of producing an efficient HCV vaccine is the high frequency of mutations and the emergence of quasi-species of the virus [35]. Until now, there has been no licensed HCV vaccine to preclude HCV infection, and the generation of an HCV vaccine poses a major challenge.

HCV is a virus belonging to the Hepacivirus genus within the family Flaviviridae [36]. HCV particles are small, enveloped, and contain a positive-stranded RNA genome of 9.6 kb in length. The genome contains a single ORF possessing well-conserved 5′ and 3′ untranslated regions that play essential roles in translating viral proteins and in replicating the viral genome [37,38]. The virion particles occur in association with lipoproteins that play roles in viral infectivity and therefore the virions circulate as lipoviroparticles within the bloodstream [39,40]. In the first stage of viral infection, the HCV particle attaches to the host cell surface, and then the HCV envelope E1 and E2 glycoproteins interact with cellular receptors [41]. Thereupon, the virus enters the host cells through clathrin-mediated endocytosis [42]. After nucleocapsid uncoating, the viral genomic RNA is delivered into the cytoplasm and then translated within the rough endoplasmic reticulum through the internal ribosome entry site (IRES) of the viral genome that in turn generates a single polyprotein. This polyprotein is cleaved into 10 polypeptides namely, the structural E1 and E2 which are the core and envelope glycoproteins, the viroporin p7, and the NS2, NS3, NS4A, NS4B, NS5A, and NS5B which are the non-structural proteins [43,44].

Madesis et al., 2010 [45] reportedly expressed the N-terminal 143 amino acids of the HCV core protein in chloroplasts of tobacco plants. In another report, the HCV core protein gene was fused C-terminally to HBsAg and codon-optimized for efficient expression in the tobacco plant (Iranian Jafarabadi-cultivar) [46] using a potato virus X-based vector (PVX). The Tomato bushy stunt virus p19 viral suppressor protein was co-expressed and this showed enhancement in the yield of the HCV core antigen. The HCV core protein was produced in transgenic canola [47] wherein soluble extracts obtained from oilseeds along with oil bodies of these canola plants proved to be immunogenic in mice. Strong IgG and T-helper 1 immune responses were elicited in addition to increased levels of IFN-gamma released from CD4+ and CD8+ cells. IL-4 cytokine levels were increased. Peptide antigen fusions using the PapMV platform have also been developed for HCV. The prolonged humoral response was generated in mice that were immunized with the PapMV-HCV E2 glycoprotein vaccine [48].

Clarke et al., 2017 [49], produced the HCV E1E2 heterodimer, a strong vaccine candidate in *Lactuca sativa* by transient agro-infiltration. In this report, both the wild-type E1E2 dimer and an E1E2 variant lacking an N-glycosylation site within the E2 protein (E1E2ΔN6) were separately expressed. When mice were administered orally with the plant material, it resulted only in weak anti-HCV serum IgM levels for both the antigens, although the E1E2ΔN6 variant elicited greater levels of secretory IgA reflective of enhanced immunogenic capabilities of the mutant. Besides, mice receiving an intramuscular prime of the HCV dimer expressed in HEK293T cells, followed by two oral boosters with the lettuce-generated E1E2 dimer developed both systemic and mucosal immunity as seen by the occurrence of HCV-specific secretory IgA in fecal extracts. The immunogenic R9 mimotope of the E2 surface glycoprotein of HCV was fused to the carriers such as the CTB [50], the surface of the Alfalfa Mosaic Virus [51], the HBsAg [32], and the CMV [52,53,54] towards developing oral vaccines in edible crops [55].

## 4. Influenza Virus Vaccines

Influenza virus (types A, B, and C) causes respiratory disease, influenza which is of major health concern to the human population [56]. Among Influenza viruses, the Influenza A H1N1, H1N1pdm09, and H3N2 viruses as well as the Victoria and Yamagata lineages of Influenza B viruses cause infections in 20–30% of children and 5–10% of adults resulting in 1 billion cases, 3–5 million severe cases, and 290,000–650,000 influenza-related respiratory deaths worldwide according to the WHO.

The influenza virus belongs to the family Orthomyxoviridae [57] and is an enveloped virus containing a segmented, negative-sense, single-stranded RNA genome. The core of the virus possesses ribonucleoprotein complexes composed of individual RNA strands, together with several monomers of nucleoprotein as well as a single copy of viral transcriptase/RNA polymerase [58,59]. The matrix protein, M1 surrounds the virus core [60] and is considered to be the main driving force behind the budding of new influenza virions through the plasma membrane of the host cell [61]. The virus envelope is supported by the M1 protein layer and is composed of lipid membrane with characteristic rod-like hemagglutinin (HA) trimers and mushroom-like neuraminidase (NA) tetramers which occur as spikes on the viral envelope [62]. The HA and NA are the principal transmembrane glycoproteins occurring in the virus envelope, of which the HA is the most abundant glycoprotein on the surface of the virus.

Influenza vaccines afford protection mainly by means of neutralizing anti-HA antibodies and due to its critical roles in establishing the early stages of influenza virus infection [63], HA becomes the primary target for the development of subunit flu vaccines. The high degree of variability of the HA antigen results in the high antigenic variability of the virus. The second HA2 subunit that forms the stalk domain is more conserved compared to the overall HA protein [64] and therefore can be employed to develop a candidate anti-influenza vaccine. The peculiarity of the influenza virus is the continued circulation of multiple epidemic strains of the virus wherein the dearth of pre-existing immunity to the emerging viral strains necessitates the manufacture of newly upgraded vaccines every year. Conventional influenza vaccines including those of the H1N1 and H5H1 viruses generated in embryonated hen’s eggs with their lengthy production time are obviously insufficient to meet the global demand should a pandemic situation arise [65,66,67,68,69,70,71]. In this context, plants are more promising as they can be established as biological factories for the mass production of low-cost recombinant influenza vaccines [72].

The production of Influenza vaccines in plants has been met with major success because the Influenza virus haemagglutinin (HA), the major factor of virus neutralization as well as the only obligate vaccine component, is expressed well and folds properly in plants. The major advantage of plant-based vaccines is the unlimited scalability of vaccine manufacture for pandemic viruses such as Influenza. In addition, they can be easily adapted to act as “rapid response” vaccines. Plant-based Influenza vaccines were mainly produced by biotechnology companies namely, Medicago USA in North Carolina, Kentucky Bioprocessing in Owensboro, Fraunhofer USA Center for Molecular Biotechnology in Delaware, the Project GreenVax consortium with partners from Texas A&M University system and G-Con from Texas [73]. Among these, Medicago produced over 10 million doses of the VLP-based H1N1 Influenza vaccine in 1 month (July, 2012) in accordance with Phase I cGMP regulations [74].

Medicago is currently the world leader in the pioneering production of plant-generated influenza VLPs. Starting with the monovalent VLP-based vaccine candidates against the pandemic influenza strains such as H7N9 [75] and H5N1 [76,77], this was quickly succeeded by a quadrivalent HA-based VLP formulation against seasonal flu which was highly successful in completed phase 1 [78], phase 2 [79] and phase 3 clinical trials (Table 1) [80]. Ward et al. 2020 [80], described two phase 3 trials wherein they showed that plant-generated HA VLPs affords superior protection against illness caused by influenza viruses in adult subjects of all ages at a degree comparable to the commercial egg-derived seasonal flu vaccines. This vaccine is presently under active review by public health authorities in different countries the world over.

Mallajosyula et al., 2014 [93] reported a TMV-based influenza HA vaccine that stimulated antibody production and better protected murine models against challenge with H1N1 influenza virus compared to the commonly administered trivalent inactivated H1N1 vaccine. The plant-produced HA-only VLPs were shown to possess a novel HA glycosylation site belonging to the H3N2 virus 2017 strain whereas the egg-adapted vaccine strain did not contain this new glycosylation site [94]. These HA-only VLPs were demonstrated to have potent immunogenicity [95,96]. Won et al., 2018 [97] showed that these HA VLPs could elicit rapid pro-inflammatory cytokine responses from human and mouse dendritic cells in vitro as well as stimulate T cells to produce antigen-specific responses. Upon immunization of mouse models, there was an accumulation of B- and T-cells as well as DCs in draining lymph nodes. Mardanova et al., 2015 [98] reported studies involving the fusion of the M2e peptide with the *Salmonella typhimurium* flagellin protein wherein they demonstrated anti-M2e antibodies in the fusion protein-vaccinated mice and showed protection against challenges with lethal doses of various strains of the virus. Considering the above, if both the M2e and HA2 domains were combined in a single candidate vaccine, it could afford a broad range of protection [99,100,101] and when further fused to an adjuvant or carrier VLP, their immunogenicity could be even more augmented [102,103].

In another study by Ward et al., 2014 [104], it was proved that HA-only VLPs generated in plants could assemble and undergo budding from plasma membranes of plant cells to generate HA-only enveloped particles. These VLPs afforded protection to influenza virulence in human subjects participating in Phase I/II clinical trials while having no negative effects due to plant-derived glycosylation on immune reactions or aggravation of plant-based allergies. Expeditious generation of GMP batches of plant-made, then newly emergent H7N9 virus HA antigen proved to protect against influenza infection in ferrets and mouse models [77]. This candidate vaccine also successfully stimulated cross-reactive humoral and cellular immune responses in human clinical trials [78]. Thus, vaccines based on plant-generated VLPs proved to be more virus-like than split vaccines produced in eggs as demonstrated by Makarkov et al., 2017 [105], wherein H1 and H5 HA VLPs made in plants closely mirrored the initial interactions of native influenza virus particles with human monocytes and macrophages.

Peptides from the influenza M2 ion channel protein were displayed on the papaya mosaic virus (PMV) which elicited antibody response capable of recognizing infected cells in addition to protecting mice from H1N1 influenza virus challenge [106,107]. Further, this formulation was combined with multimerized nucleoprotein nanoparticles which afforded protection to challenges by both H1N1 and H3N2 influenza virus strains in murine models [108]. Besides, PapMV VLPs are inherently capable of adjuvant activity which dramatically enhanced immunogenicity and provided 100% protection against the WSN/33 influenza virus strain challenge [109]. Peptides from the influenza virus M1 matrix protein and the nucleocapsid protein were displayed on the surface of PapMV particles which activated B cells and led to the expansion of antigen-specific T cells in vitro and immunization of mice with this formulation showed elicitation of antigen-specific CD8+ T cells [110,111,112].

In yet another study [113], the influenza virus M2e peptide was fused into the P2 loop of the Hepatitis E Virus ORF2 protein after the Gly556 residue and expressed in *N. benthamiana* plants. This resulted in an HEV capsid protein yield of 10% of the total amount of the soluble protein. However, only the capsid protein-containing HEV 100–610 amino acids and the chimeric M2e HEV 110–610 protein could spontaneously assemble into higher-order structures. In particular, the latter chimeric VLPs assembled into 22–36 nm particles which recognized anti-M2e antibodies.

Blokhina et al., 2020 [114], demonstrated transient expression of *S. typhimurium* flagellin protein fused to the conserved sequence (76–130 amino acids) of the HA2 subunit (belonging to the first phylogenetic cluster of influenza A viruses) and the M2e peptide fused as four copies in tandem. This recombinant hybrid protein was produced in *N. benthamiana* plants using a PVX-based self-replicating vector with a yield of up to 300 ug per gram of fresh leaf weight. When the purified fusion protein was intranasally administered to mice, it induced enhanced levels of anti-M2e serum antibodies and afforded strong protection against lethal doses of influenza A virus strain A/Aichi/2/68(H3N2). In another report, *N. benthamiana* was used to express trimeric H7 hemagglutinin conjugated to the surface of nanodiamond particles [96,115]. In mouse models, two to three doses of the immunogen containing the medley of trimeric H7 protein and the nanodiamond demonstrated a considerably potent H7-specific IgG immune response. The H5N1 A/Indonesia/5/05 HA VLP candidate vaccine from Medicago was the first reported case of administration of a plant-based VLP vaccine to humans [75]. In human trials, the H5 HA VLPs administered along with the alum adjuvant in 2 doses afforded immunity to 96% of healthy individuals used in the study. Smith et al., 2020 [116] demonstrated that an H6-based influenza A VLP vaccine generated transiently in *N. benthamiana* (Figure 1) afforded protective immunity in chickens challenged with the Influenza A H6N2 virus.

As discussed above, plant-derived Influenza viruses and HA-based plant vaccines show great promise in treating influenza infections. However, other Influenza virus proteins such as the M1 matrix protein, the NA neuraminidase, and the NP nucleoprotein that play important roles in immunity to the virus remain to be explored as potential plant-based vaccine candidates.

## 5. Papillomavrius Vaccines

Of the several cancers affecting women, cervical cancers rank the fourth with over ~570,000 new cases recorded in 2018 in under-developed countries where there occur 84% of the global cervical cancer cases. At present, there are three commercially available VLP-based prophylactic vaccines against HPV, namely, Cervarix., Gardasil^®^, and Gardasil9^®^. The Cervarix affords protection against infections by HPV16 and 18, while Gardasil^®^ is specific to HPV 6, 11, 16 and 18 [117] and Gardasil9^®^ protects against 9 HPV types namely, HPV 6, 11, 16, 18, 31, 33, 45, 52, and 58 [118]. All of these vaccines have been deemed highly efficacious and safe. However, the cost of production of these vaccines is prohibitively high thus precluding their widespread use in developing countries with a higher incidence of HPV infection [119,120].

The HPV virion contains an icosahedral capsid of 55 nm diameter, composed of 72 capsomeres encapsidating a closed circular, double-stranded DNA genome [121]. One of the DNA strands codes for proteins classified as early (E) and late (L) proteins [122]. Out of these, proteins E1, E2, E4, E5, E6, E7 are non-structural polypeptides involved in functions such as transcription, replication, transformation, and viral escape. The structural proteins L1 and L2 form the virus capsid [123]. Of these, the L1 polypeptide capsomeres by themselves can form VLPs that exhibit morphological identity with the real, intact viral capsid [124] except in lacking viral DNA or RNA. These L1 VLPs are entirely non-infectious as well as non-oncogenic and are currently employed with great success as HPV vaccine candidates [125]. Despite being highly immunogenic [126,127], well-tolerated and greatly successful in protecting against different HPV types, the above VLP-based HPV vaccines incur high costs while requiring a continuous cold chain and needing to be administered by intramuscular injection. Additionally, vaccine production systems based on fermenters make the vaccine highly expensive. Therefore, these factors preclude the use of the above VLP-based HPV vaccines in developing countries with over 85% incidence of disease due to cervical cancer [128]. This necessitates the generation of economical second-generation vaccines with improved ease of administration while providing long-term immunity [129]. In this context, plants offer a cost-efficient alternative for the generation of HPV vaccines due to their high scalability, robust yield, safety, and innate ability to incorporate post-translational modifications as well as facilitate foreign protein assembly [130,131].

Chabeda et al., 2019 [132] demonstrated the success of HPV vaccine candidates generated from HPV L2 protein, in particular the peptides 17–36, 56–81, 65–81, and 108–120 which are conserved to a great extent across many HPV types. These L2 peptides each were substituted into the HPV-16 L1 protein DE loop at position 131 or the L1 C-terminus at position 431 to produce L1:L2 chimaeras derived from HPV-16. When all of these chimaeras were expressed transiently in *N. benthamiana* and used to immunize mice, they showed cross-neutralizing antibodies to other HPV types such as the HPV-11, 18, and 58 in addition to HPV-16.

Naupu et al., 2020 [133] generated a trivalent vaccine candidate against HPV 35, 52, and 58 by transient expression in *N. benthamiana* wherein they demonstrated successful HPV L1-specific humoral immune response at levels equivalent to that elicited by the HPV Gardasil^®^ vaccine. The HPV16 L1 protein was produced in *Nicotiana tabacum* chloroplasts [134] wherein protein expression was driven by an ethanol-inducible promoter. This led to the accumulation of the L1 protein at levels up to 3 μg/mg of fresh plant material. This study is a good example of the inducible expression of transgenes in plants.

Salyaev et al., 2019 [135] reported the expression of the highly antigenic L1 capsid epitopes of HPV16, HPV18, HPV31, and HPV45 types wherein the Cucumber mosaic virus replicase gene was inserted into the construct carrying the HPV epitopes. This resulted in the enhanced generation of these antigenic HPV proteins to levels as high as 25–27 μg/mg of the total soluble protein. When compared with the quadrivalent Gardasil vaccine against the HPV types 16, 18, 6, and 11, this CMV-based vaccine methodology generated higher amounts of the HPV antigenic proteins. Thus, the use of viral *rdrp* structural elements as well as the associated regulatory genes which act as suppressors of RNA silencing enabled a several-fold enhancement of production of the HPV antigens. SAPKQ, a nontoxic form of the saporin protein obtained from Saponaria officinalis was fused to the HPV16 E7 protein followed by expression of this candidate vaccine in hairy root cultures of tomato plants using a recombinant plant expression vector [136]. When mice were immunized with this vaccine formulation, it demonstrated a strong immune response against tumors thus showing anticancer activity. Therefore, hairy root cultures can be used as low-cost biofactories to develop therapeutic HPV vaccines. Yazdani et al. 2019 [137] reported the generation of the grapevine fanleaf virus VLPs displaying the L2 epitope of the HPV. The minor capsid protein L2 of HPV is a favored candidate to develop broadly protective HPV Vaccines although the L2 protein by itself is weakly immunogenic. The L2 protein was displayed on the surface of VLPs derived from HBcAg or was fused genetically to an immunoglobulin molecule capable of generating recombinant immune complexes (RIC) [138]. Both of the above vaccine candidates showed a robust immune response and were even more potent when administered together. A high antibody titer against the L2 protein was observed concomitant with virus neutralization.

From the above, it is evident that plants could serve as a really useful platform for the generation of both prophylactic and therapeutic HPV vaccines. The VLP-based HPV candidate therapeutic vaccines being the gold standard can be easily made at enhanced yields through transient expression. Further research is underway to make plant-based HPV vaccines that are both prophylactic and therapeutic.

## 6. HIV Vaccines

More than about 38 million people are living with HIV infection wherein about 24 million are undergoing treatment with antiretroviral therapy (ART) [139]. Over 700,000 of the human population are infected with HIV every year and many of them die from chronic AIDS. About 2/3rds of the infected people are from Africa. The need for a successful prophylactic vaccine against HIV is dire because, in HIV endemic areas, people have little or no access to ART due to low socioeconomic status. ART requires lifelong intake with several side effects. The USA had executed a phase I clinical trial of a prospective candidate vaccine way back in 1987. Presently, over 30 HIV candidate vaccines have entered into phase I/II clinical trials performed principally in Europe and the USA.

HIV is a lentivirus, a subgroup of the family retroviruses, and contains a genome of two single-stranded RNAs as well as many viral proteins encapsidated within an enveloped viral capsid. There are two HIV types identified, HIV-1 and HIV-2, depending on their genetic composition and the several viral antigens. Nevertheless, the majority of the world’s AIDS pandemic is caused by HIV-1 and its subtypes. The viral envelope is composed of trimers of gp120 and gp41 heterodimers that are linked by non-covalent interactions. Thus far, over 12 HIV-1 subtypes and hundreds of recombinant forms of HIV-1 are in circulation [140]. The high mutation rate of the gp120 HIV envelope protein has serious implications for immunity against HIV [141]. Not only is the HIV envelope rapidly variable, but the genome of the virus also mutates at a high rate (~1–10 mutations per virus replication cycle). Additionally, there is extensive conformational flexibility and glycan coverage. Further, the virus has evolved a multitude of mechanisms to evade the host’s neutralizing antibodies [142]. Generation of HIV polyvalent vaccines capable of recognizing conserved regions on the viral envelope can help overcome the high degree of changes in the viral envelope [143].

Recently, an anti-HIV-1 protein consisting of bispecific broadly neutralizing antibody-lectin fusion has been transiently in *N. benthamiana* [144]. Human monoclonal antibody 2G12 capable of neutralizing HIV by recognizing carbohydrate epitopes on the HIV surface has been produced in tobacco and is presently approved for phase I human clinical trials [145]. A gp120-based multi-epitopic chimaeric protein C4(V3)6 was expressed in plants such as lettuce [146] and tobacco [147] which exhibited strong immune responses against HIV [148]. This provided the C4(V3)6 as a prospective polyvalent vaccine against HIV. Similarly, the moss plant called *Physcomitrella patens* was used to generate a chimaeric HIV protein derived from the epitopes of gp41 and gp120 resulting in a vaccine, pol-HIV [137,149]. The chimaeric protein extracts induced antibody responses in mice, thus proving the moss plants as an expression platform for the production of HIV vaccines and antigens. In another study, the epitope ELDKWA capable of capturing the HIV neutralizing antibody 2F5 was designed to be carried on the DsRed fluorescent protein making the DFE fusion polypeptide which was subsequently expressed in transgenic tobacco [150]. The yield of this protein was shown to be about 24 mg/kg plant material with as high as 90% purity. This has the potential to be used as a cost-effective alternative for the generation and purification of idiotype-specific HIV mAbs. Rubio-Infante et al., 2015 [151], report the expression of a tobacco-produced multi-epitopic immunogenic HIV protein which when orally administered to mice, induced T-helper cell responses and g-IFN production.

HIV neutralizing proteins including the HIV Mab 2G12, cyanovirin-N, and griffithsin (an anti-HIV algae-derived lectin molecule) were recently generated in rice endosperm [152], which together neutralized HIV by interaction with gp120. When these proteins were expressed in plants such as tobacco and lettuce, they neutralized the virus more potently when compared to the conventionally expressed cocktail. A TMV-based griffithsin expression vector system has been used to produce griffithsin, a highly efficacious HIV entry inhibitor in tobacco with yields as high as over 1 g/kg fresh leaf material that also exhibited a strong ability to bind to gp120 [153]. This subsequently enabled production capacity of as much as 20 kg of griffithsin per year [154] at a much lower cost than the equivalent mammalian cell-based production systems [155]. A lectin called cyanovirin has been synthesized in tobacco [156]. The banana lectin possesses anti-HIV replication activity [157] by specifically targeting the glycosylated gp120 in the region of its dense mannose glycans [158]. Recently, HIV Env gp140 antigens have been synthesized by transient expression in *N. benthamiana* [159]. The green alga, Chlamydomonas has been used to express the p24 HIV antigen at levels as high as 0.25% of the total cellular protein [160].

Medicago Inc. reported high-level plant production of a chimeric form of the HIV Env protein fused to the Influenza HA transmembrane and cytoplasmic tail domains which successfully budded into the HIV VLPs even without the core or matrix proteins [161,162]. This provides a truly novel source of the Env antigen with augmented immunogenicity and could be of great use as a booster vaccine for the heterogeneous prime-boost vaccination schemes. A Gag-based VLP-like molecule was generated in transgenic *N. benthamiana* [163] and transplastomic *N. tabacum* [164]. Among these, the transgenic *N. benthamiana* showed stable expression of the Gag protein which interacted with the co-expressed gp41 (a part of the external region of the Env protein), to generate enveloped VLPs. These VLPs elicited a robust immune response to the Gag protein in mice. Porta et al., 1994 [165], engineered a 22 amino acid epitope from the HIV-1 gp41 to be displayed on the surface of CPMV which showed neutralizing response against three strains of HIV-1 in mice that were parenterally administered with the vaccine formulation [166,167]. Sera from these mice were shown to contain HIV-1-specific antibodies capable of recognizing two distinct epitopes on the gp41 peptide, one of them neutralizing and the other, non-neutralizing [168]. The HIV p24 capsid protein was expressed in transgenic tobacco (Zhang et al., 2002) [169]. The Tat protein was produced in spinach which showed a priming effect upon administration after primary injection with a Tat-specific DNA vaccine (Karasev et al., 2005) [170].

## 7. SARS-CoV-2 Vaccines

It has been reported that as of January 2021, the COVID-19 coronavirus disease has spread to almost all the countries of the world affecting nearly 84 million individuals and causing 1.8 million deaths. The first case of Covid19-induced hospitalization was reported in Wuhan, China in December 2019, WHO, 2019, [171]. In comparison with the H1N1 Influenza virus that has a 0.02% mortality rate, Covid-19 has a higher (3%) mortality rate [172]. Coronaviruses belong to the family Coronaviridae. Before the emergence of SARS-CoV-2, betacoronaviruses SARS-CoV and MERS-CoV were reported to be highly pathogenic [173], Coronavirus COVID-19 Global Cases by the Center for Systems Science and Engineering (CSSE) at Johns Hopkins University, 2020. The SARS-CoV-2 principally infects the respiratory tract, gastrointestinal tract, and even the brain in several instances. The virus in the early stages of infection causes cough, fever, fatigue, vomiting, loss of smell/taste, and dyspnea [174,175,176,177,178] before manifesting as full-blown infection within the body and causing serious diseases such as pneumonia, lung and multi-organ failure and even death. The SARS-CoV-2 virus spreads through the population by means of respiratory secretions from the infected individual during sneezing, coughing, or talking. Droplets emerging from the infected person infect neighboring individuals when they come in contact with the latter’s mucous membranes [179]. These droplets emerge from the infected person within a range of 2 m and are additionally likely to contaminate surfaces in the vicinity which enables virus spread when an individual touches these surfaces followed by touching the nose, mouth, or eyes. Although the virus can prevail and spread through asymptomatic individuals, symptomatic patients are most contagious [180].

The SARS-CoV-2 is an enveloped virus consisting of a 29,881 nucleotide, positive-sense, single-stranded linear RNA genome [181,182,183] that codes for at least four main structural proteins including the spike glycoprotein (S), the envelope protein (E), the membrane protein (M) and the nucleocapsid protein (N). The S protein forms homotrimers that radiate from the surface of the virus, thus presenting a typical crown-like appearance that is found in all coronaviruses. The S protein enables virus entry into host cells by recognizing and interacting with the Angiotensin-converting Enzyme 2 (ACE2) receptor using its Receptor-binding Domain (RBD) that is part of the S1 subunit (Figure 2) [184].

The S protein is the major antigen capable of eliciting robust immune responses [185]. The S protein undergoes proteolytic cleavage into the S1 subunit composed of 685 amino acids and the membrane-spanning S2 subunit composed of 588 amino acids. The S2 is up to 99% conserved among the CoV families, while the S1 protein displays only 70% identity to other strains of human CoV wherein the differences occur mainly in the RBD [186]. Since the most effective approach to control viral infection is by precluding virus entry into the cell, the S protein is the most appealing vaccine candidate capable of eliciting neutralizing antibody responses or cross-presentation/antibody-dependent cell-mediated cytotoxicity (ADCC) to generate protective immunity at the cellular level [187].

Even while social distancing, masking, and contact tracing practices can slow down the spread of the Covid-19 virus, nevertheless it seems to be too infectious to be eliminated simply by following these strategies in addition to the recent emergence of more infective S protein variants [184]. Therefore a potent vaccine (prophylactic/therapeutic) is imperative to facilitate a normal return to human social interaction. The worldwide venture to produce a suitable vaccine for Covid-19 has become successful. At present, over a dozen vaccines have been authorized for use and are being administered currently across the world while several more are still being developed.

For SARS-CoV-2, the conventional strategy of using attenuated or inactivated strains of the virus has several caveats such as the lengthy timeframe required to generate enough vaccine, reactogenicity, antibody-dependent enhancement of infection in addition to the risk of virulence reacquired by reversion mutations and other safety issues [188,189,190]. Compared to this, an easier and safer option would be to generate subunit vaccines derived from the expression of individual SARS-CoV-2 antigens or VLPs composed of SARS-CoV-2 antigens presented in multiple copies arrayed on the VLP surface [191]. Moderna and Pfizer/BioNTech encapsulate their mRNA vaccines within lipid nanoparticles (LNPs) and the Astrazeneca/University of Oxford and CanSino incorporate antigen-encoding genetic sequences within the DNA of Adenovirus [192,193,194,195]. On the other hand, Novavax technology uses the expression and display of recombinant S proteins on their proprietary VLP nanoparticles [196]. The current situation of the rampant spread of SARS-CoV-2 necessitates the production of low-cost, rapidly produced stable vaccines without any cold chain requirements and usable even in developing countries, and therefore plants provide a viable platform for the generation of SARS-CoV-2 vaccine. Formulating epitope-based vaccines is also a better option to reduce the risk of disease enhancement.

British American Tobacco (BAT) and its US biotechnology subsidiary, the Kentucky Bio-Processing (KBP) are currently generating a plant-based Covid-19 vaccine in tobacco wherein they have expressed SARS-CoV-2 protein subunits. Specifically, they have used the entire S1 polypeptide or the smaller RBD within the S1 as vaccine candidates. Thus far, the vaccine has been shown to elicit a positive immune response in pre-clinical testing [197] and is progressing into Phase I/2 human clinical trials (Table 1) [198]. This vaccine candidate is capable of inducing an efficient immune response in only a single dose and has been shown to the stable at room temperature (BAT 2020). The S1 polypeptide is highly glycosylated and these glycans are formed of a medley of complex and high-mannose components making it essential to generate the whole S1 and RBD with signal peptides at their N-termini to secrete the proteins into the endomembrane system [199]. BAT has the potential to manufacture as high as 1–3 million of the Covid19 doses per week as they have already produced 10 million vaccine doses for Influenza in a month and the Ebola vaccine using the same plant-based strategy [200].

The Canadian biopharmaceutical company, Medicago is pioneering the development of the anti-SARS-CoV-2 plant-based vaccine wherein they utilized their know-how with plant-generated influenza VLPs to formulate the VLP-based SARS-CoV-2 vaccine. They generated VLPs of the SARS-CoV-2 by the insertion of the gene sequence of the SARS-CoV-2 spike protein into Agrobacterium followed by the infection of *N. benthamiana* plants with the engineered Agrobacterium [201]. These plants developed SARS-CoV-2 VLPs composed of the spike protein and the plant lipid membrane. These VLPs are similar to the actual virus in size and shape but are devoid of the virus genetic material and therefore are non-infectious. Specifically, their VLPs consist of a modified version of the SARS-CoV-2 S protein having stabilizing point mutations (R667G, R668S and R670S 217 substitutions at the S1/S2 cleavage site), a plant-specific signal peptide in place of the native sequence as well as the transmembrane domain and the cytoplasmic tail of S protein supplanted with equivalent sequences sourced from influenza H5 A/Indonesia/5/2005 to increase VLP assembly and budding [202]. This vaccine formulation was demonstrated to be successful in phase I human clinical trials and was found to be highly immunogenic, safe, and well-tolerated. This VLP is currently in phase 2–3 clinical trials and is estimated to manufacture this candidate vaccine at the rate of 10 million doses per month [203,204].

In Canada, Suncor and the University of Western Ontario are producing Covid19 diagnostic test kits using algae as a production platform to generate the Covid19 spike protein [205]. Through this, algae are ideal biofactories as they are easy to grow and can be readily engineered to express viral proteins. iBio (Bryan, TX, USA) is expressing a VLP-based anti-SARS-CoV-2 vaccine in tobacco derived from their proprietary FastPharming scheme [iBio (2020), [172]. The iBio American company is currently developing a plant-made anti-SARS-CoV-2 subunit vaccine candidate [206] using segments of the major surface glycoprotein, the spike (S) protein in fusion with lichenase (LicKM), a carrier protein obtained from Clostridium thermocellum b-1,3-1,4-glucanase. Cape Bio Pharms (CBP), a South African company has produced diagnostic reagents for SARS-CoV-2 in plants which can be used in virus diagnosis [207]. It has generated the virus spike S1 protein containing different regions of the glycoprotein attached to various fusion proteins. Additionally, this company is working in collaboration with antibody manufacturers to express antibodies against these proteins in plants [207]. The research group headed by Nicole Steinmetz at the University of California, San Diego has displayed the B- and T-cell epitopes of the SARS-CoV-2 S protein on the surface of the icosahedral cowpea mosaic virus [208]. This formulation can be administered using an implanted microneedle technology that incorporates VLP vaccines in the skin that is capable of eliciting anti-SARS-CoV-2 immune response [209]. This CPMV-based technology has been also used to produce Covid19 diagnostic testing kits with improved accuracy. These kits are highly beneficial as they are inexpensive to manufacture, are highly stable, and can be stored at room temperature for lengthy time periods which can be of value in resource-poor environments.

Another couple of research groups based in Toronto, Canada has engineered a novel method to combat Covid19 using a synthetic peptide capable of binding the viral deubiquitinase (DUB) and is displayed by a plant virus. This investigation began first by studying the role of the viral protease encoded by the ORF1a of the related Middle East Respiratory Syndrome (MERS) virus. This protease possesses a deubiquitinase activity that protects the virus from ubiquitin-mediated degradation by the cellular innate immune system. An 80 amino acid synthetic peptide named the ubiquitin variant (UbV) was generated by phage library display design and this was demonstrated to bind tightly to the deubiquitinase of the MERS virus at its ubiquitin-binding site, effectively blocking its deubiquitinase and protease activity. Consequently, the synthetic UbV peptide blocked the MERS virus infection when tested in a human cell line wherein a lentivirus vector was used for cell entry [210]. Following this initial study, another UbV analog capable of binding the SARS-CoV-2 DUB has been generated for use in controlling SARS-CoV-2 infection [211].

Currently, studies are underway to investigate if the above virus-peptide fusions can assemble into VLPs. Earlier work has demonstrated that PaMV can enter human cells via the cytoskeletal protein, vimentin [184]. Thus, it is expected that these PaMV-derived virus nanoparticles loaded with the UbVs of these viruses can enter the cells and effectively inhibit virus infection. Additionally, these PaMV VNPs were shown initially to enter the lung epithelial cells when administered as a nasal aerosol spray. Therefore, potentially these UbV-carrying VLPs can be loaded into an inhaler to treat the nasal cavity and the lungs of infected SARS-CoV-2 patients. In this context, it has been shown that the genetically engineered Bean yellow dwarf virus, a Geminivirus has yielded large amounts of biopharmaceutical proteins in relatively short time periods [212].

Furthermore, another unique anti-SARS-CoV-2 synthetic antibody that was genetically engineered from a phage display library is also currently under investigation in a geminivirus vector system [213]. Likely targets could be the spike protein, the entire nucleocapsid, the membrane, the envelope, the viral RNA polymerase as well as the 3-chymotrypsin-like protease (3CLpro). The 3CLpro plays an important role in the biology of SARS-CoV-2 by cleaving the virus polyprotein at 11 distinct sites which generate non-structural proteins crucial for viral replication.

Other pharmaceutical companies that are involved in the race for plant-based SARS-CoV-2 vaccines include Greenovation Biopharmaceuticals, Nomad, Ventria, and Protalix [172]. Among academic institutions interested in developing the Covid-19 vaccine, the Laval University Infectious Disease Research Center in Quebec, Canada is collaborating with Medicago to develop anti-SARS-CoV-2 therapeutic antibodies. Many universities and institutes in countries such as the UK, USA, Germany, South Korea, Thailand, South Africa, and Mexico are interested in performing anti-Covid-19 molecular pharming [172]. VLPs by virtue of their inherent lack of capacity for replication and deconstructed viral vectors in addition to the use of the *N. benthamiana* as host plant hold great promise towards the generation of Covid-19 vaccines. Long-term objectives include the generation of transplastomic lines and edible plants with the nuclear transformation that can be used as oral vaccine booster shots to enable mucosal immunity. The real test for the development of effective plant-derived Covid-19 vaccines would be their resilience in large-scale clinical trials to validate their efficacy and safety while fulfilling the regulatory agency requirements. The existing precedence of other biopharmaceuticals such as plant-based vaccines against Influenza is an encouraging factor. Even as the Covid19 spreads at pandemic proportions, plant-based vaccines show increasing promise to produce easy to administer, low-cost, safe, and efficacious vaccines against this deadly virus and the coming few months would be critical in realizing the full potential of this emerging technology. The ideal plant-based vaccination scheme would involve a combination of parenteral administration of purified transiently expressed injectable plant-based vaccines followed by oral boosters with plant biomass containing the vaccine antigen.

## 8. Zika Virus Vaccines

More than 69 countries have reported Zika virus infections [214] between 2015 and 2017. In 2016, the World Health Organization declared the Zika virus infections as a Public Health Emergency of International Concern. To date, about 45 candidate vaccines have been tested in non-clinical investigations, of which there is at least one in phase II human clinical trials and many are in phase I trials [215,216,217]. Zika virus infection manifests itself as a self-limiting illness symptomatized by rash, fever, myalgia, and headache. In its most severe form, it causes extreme abnormalities such as microcephaly in the fetus and leads to Guillain–Barre syndrome in adults [14,218,219]. At present, there exists no therapeutics or vaccines against this virus and there is a compelling need to generate safe and efficacious vaccines to preclude Zika virus infection especially in pregnant women.

The Zika virus is a Flavivirus and is related closely to the 4 serotypes of Japanese encephalitis virus (JEV), tick-borne encephalitis virus (TBEV), dengue virus (DENV), yellow fever virus (YFV), and the West Nile virus (WNV) [208]. As in other flaviviruses, the Zika virus encodes the Envelope glycoprotein (zE) which contains the EDI, EDII, and EDIII ectodomains. The zE protein ectodomains perform the functions of recognizing and attaching the virus to cellular receptors, followed by membrane fusion enabling virus entry in addition to mediating virus assembly [14,220]. Amongst these ectodomains, the zDIII domain is the principal candidate for a subunit vaccine as it is highly conserved [221] and has the potential to induce robust neutralizing antibodies.

One of the potential threats of the flavivirus vaccines is the induction of antibody-dependent enhancement (ADE) wherein non-neutralizing antibodies generated in response to vaccination or infection by one of the flaviviruses cross-react with another infecting virus to form complexes. These complexes recognize the cellular Fc-c receptors or the complement-associated receptors and are ingested by myeloid cells. This results in enhancement of the virus infection [222] as the antibodies involved in complexation fail to neutralize the virus. Thus, the ADE effect causes the Zika virus E protein fusion loop to augment dengue virus infection. On the other hand, the Zika EDIII domain (ZE3) induces type-specific neutralizing antibodies that are not complicated by the ADE peptide but are weakly immunogenic and for this reason, it has been employed to fuse with the RIC (reactive immune complexes) antibody which demonstrably enhances the B- and T-cell reactions even in the absence of any adjuvant [223,224,225,226]. Diamos et al., 2020 [214], have demonstrated a high level neutralizing immune response in mice treated with the correctly assembled plant-derived RICs or VLPs expressing the Zika virus ZE3 antigen. Upon codelivery of both the RICs and the VLPs, there was a synergistic increase in Zika virus neutralization and antibody levels specific to ZE3. Moreover, when hepatitis B core (HBcAg) VLPs displaying ZE3 peptide were expressed in plants, it resulted in similar enhancements in antibody titers and neutralization responses [14]. There was also a notable increase in IFN-gamma levels which implies potent cellular immune responses particularly, the Th1 or Th1/Th2 responses vital to virus neutralization and germane to the prevention and treatment of viral infections [14]. Cabral-Miranda et al., 2019 [227] reported a CMV-based vaccine displaying the EDIII Zika virus envelope protein which elicited increased levels of antibodies specific to this protein while enabling neutralization of the zika virus without enhancement of dengue virus infection.

Diamos et al., 2020b [228] reported the expression of a broadly neutralizing chimeric anti-flavivirus murine antibody 2A10G6, in which its variable regions were codon-optimized and subjected to fusion with human IgG1 antibody. This chimaeric antibody was expressed in plants at high yields (1.5 g/kg leave tissue) involving facile single-step purification. This antibody could recognize the E protein of the Zika virus and could robustly neutralize the Zika virus. Yang et al., 2017 [14] used the MagnICON vector to transiently express the HBcAg-zDIII fusion protein as VLPs in *N. benthamiana* using Agrobacterium infiltration (Figure 3). This was shown to afford humoral and cell-mediated immunity in mice while eliciting antibodies that did not enhance dengue virus infection. This proved to be a low-cost, safe alternative for generating immunity to the Zika virus.

## 9. Other Plant-Based Vaccines

The tropical disease, malaria is caused by protozoan parasites that infect a large number of the human population with over 219 million new cases reported each year. Despite the existence of several prevention treatments, there are no efficient, licensed, widely usable anti-malarial vaccines. Plant-based vaccines provide novel platforms for generating reliable, safe, and low-cost treatments against malaria. Chichester et al., 2018 [229] recently reported the transient expression of VLPs composed of the *P. falciparum* surface (pfs25) antigen and the alfalfa mosaic virus coat protein in *N. benthamiana*. This malaria vaccine candidate is safe and non-toxic in Phase I clinical trials. The 19 kD C-terminal fragment of *P. falciparum* MSP119 (PfMSP119) was first expressed in plants via stable transformation [230]. This antigen by itself provided protection against malarial infections in monkeys [231] and mice [232] and therefore is considered as a prime vaccine candidate against blood phases of malaria. A chimeric protein composed of both MSP119 and AMA1 domain III elicited antibody reactions at higher levels than when expressed as individual components [233]. Moreover, anti-PfCP-2.9 sera generated in rhesus monkeys and rabbits inhibited the growth of *P. falciparum* lines, 3D7, and FCC1/HN in vitro. This inhibition was found to be dependent on the elicitation of antibodies to the chimeric protein and their disulfide bond enabled conformations.

The *P. knowlesi* AMA1 antigen in its purified state is potently immunogenic in rabbits when administered along with CoVaccine HT, an adjuvant [234]. Transgenic tobacco lines that expressed an immunoreactive PyMSP4/5 were developed by Wang et al. [235] and these induced antigen-specific antisera in mice. A cocktail of antigens from different stages of malarial infection could be more effective as a multicomponent multi-stage vaccine. A physical mixture of two recombinant polypeptides, MSP-119 and F2 (the receptor-interacting F2 domain of the erythrocyte binding antigen, EBA175) was shown to induce a weak anti-MSP1 immune response [236] wherein the EBA175 acts as a high-affinity ligand that interacts with sialic acid residues of glycophorin A occurring on the surface of red cells to mediate invasion. An equal mixture of four *P. falciparum* recombinant antigens derived from namely, the pre-erythrocytic (*Pf*CSP_TSR, *Pf*CelTos, and *Pf*TRAP_TSR), the blood (*Pf*AMA1, *Pf*MSP1-19_EGF1, *Pf*MSP4_EGF, *Pf*MSP8_EGF1, *Pf*MSP8_EGF2, and *Pf*MSP3), and the sexual stages (*Pf*s25 and *Pf*s230) were expressed transiently in *N. benthamiana* [237]. Both MSP1 and AMA1 are promising vaccine antigens against malaria as they are shown to be required for merozoite-mediated invasion of erythrocytes. Whereas the MSP1 has been found to occur along the surface of the merozoites, the AMA1 has been located in apical organelles from where it is discharged onto the surface of the merozoites during or just before erythrocyte invasion [238,239]. Both MSP1 and AMA1 have also been shown to induce protective immune responses to infection by the malarial parasite in non-human primate model systems and rodents [240,241,242,243].

Milan-Noris et al., 2020 [244] expressed a Malchloroplast candidate vaccine formed of segments of the two epitopes MSP1 and AMA1 of *P. falciparum* along with the *Taenia solium* GK1 peptide adjuvant that was synthesized in tobacco chloroplasts. This Malchloroplast vaccine candidate was shown to induce antigen-specific humoral responses in mice upon subcutaneous administration. This vaccine was also capable of recognition by antibodies of patients having *P. falciparum* malaria and was also found to be immunogenic in mice. Therefore, this investigation afforded proof of concept for a dependable plant-derived antimalarial candidate subunit vaccine.

Mycobacterium tuberculosis causes tuberculosis (TB), a deadly infectious disease that is spread worldwide. An efficient vaccine is considered wanting especially in developing countries with a high prevalence of the disease. The N-glycosylated antigen 85A (G-Ag85A), a well-studied candidate vaccine antigen was expressed in *N. benthamiana* [245]. This antigen was shown to induce a more potent IFN-gamma response compared to its nonglycosylated counterpart (NG-Ag85A) generated in *E. coli* and was recognized well by the immune system of the host during tuberculosis infection. It also provided moderately increased long-term protection and balanced multifunctional Th1 immune responses along with sustained IFN-gamma response. This proved that G-Ag85A could be a good Mtb subunit vaccine antigen. The M. tuberculosis 6 kD early secretory antigenic target (ESAT-6) was produced by Saba et al., 2020 [246] in *Brassica oleracea var. italica* (broccoli) using Agrobacterium-mediated transformation to enable oral delivery of the antigen. This antigen elicited a humoral immune response in mice upon oral and subcutaneous administration. This expression of the Mtb antigen in edible plants could help in developing low-cost oral delivery of the TB vaccine. In yet another study [247], the ESAT-6 antigen of Mtb was expressed in *N. tabacum* chloroplasts by an inducible T7 promoter. Induced plants accumulated up to 1.2% of the total soluble protein.

Dengue fever is caused by the dengue virus that is endemic to over 120 countries accounting for 3.9 billion of the populace at risk of dengue virus infections. The proper treatment of this disease, in the absence of an effective vaccine against the four dengue virus serotypes, necessitates the development of rapid and efficient diagnostic methods for controlling the spread of infection. Detection of anti-dengue antibodies is complicated by the absence of large-scale production of the dengue virus non-structural 1 (NS-1) protein to be used in capturing antibodies from the blood serum of dengue patients. Xisto et al., 2020 [248] expressed the NS1 protein of the dengue virus serotype 2 (NS1DENV2) using transgenic Arabidopsis thaliana wherein 203 mg of the recombinant NS1 protein was obtained per gram of fresh leaf material. This plant-produced antigen exhibited high specificity and sensitivity to both IgM and IgG. This study validates the employment of plants as a valuable means for the efficient, large-scale expression of the dengue virus protein for the diagnosis of dengue infections.

The only licensed anti-dengue vaccine, Dengvaxia may not be sufficiently safe in seronegative and young patients. Therefore, the development of safe, efficacious vaccinations against dengue is called for. Ponndorf et al., 2021 [249] report the transient expression and assembly of DENV VLPs in *N. benthamiana* wherein the DENV structural proteins (SP) and a truncated form of the non-structural (NSPs) lacking the NS5 (that codes for the virus RdRp) were co-expressed. These VLPs were comparable to those expressed in mammalian cells in terms of size and appearance. The plant-produced DENV1-SP + NSP VLPs elicited better antibody responses in mice when compared to that of the response generated by the DENV-E domain III expressed in bluetongue virus core-like particles and a DENV-E domain III subunit. This study supports the idea of the optimal use of VLPs in generating successful vaccine candidates against enveloped viruses.

## 10. Plant-Based Therapeutic Antibodies

The development of therapeutic antibodies forms a major sector of the biopharmaceutical market worldwide [250]. Plant-based antibody production has become the system of choice as they dispense with the need for expensive infrastructures such as bioreactors or cell-culture stations or sterile environments while enabling complex posttranslational modifications that are vital to the function of these antibodies [251]. They are also inherently safe as they do not carry animal pathogens unlike mammalian systems and are amenable to large-scale production at low costs making them suitable for conveying prophylactics and therapeutics to developing countries [252,253,254,255]. Furthermore, plant genetic engineering has enabled the development of “tailored” glycans which are more human-like and homogenous compared to mammalian systems [256]. This is of great benefit in the development of plant-based monoclonal antibodies wherein proper glycosylation is vital for antibody function and stability [257]. Plant-based antibodies with higher potency and augmented ability to bind immune receptors compared to mammalian-based antibodies have been produced by the removal of plant-specific endogenous b1,2-linked xylose and a1,3-linked fucose sugars [258,259]. Significantly, plants have been engineered to code for the complete human sialyation pathway [260,261].

Glycoengineered plants have been used to produce antibodies crucial in the treatment of Ebola virus infections in humans and rhesus macaques [262,263,264]. Ma et al., 2015 [133] reported the pioneering human clinical trials using plant-based antibodies. Moreover, plant-made therapeutic antibodies showing high-level expression, safety, and efficacy have been generated against infections due to the West Nile virus [265], the dengue virus [266], and the chikungunya virus [267].

Of great importance among plant expression systems, is the bean yellow dwarf geminivirus (BeYDV) system wherein the target gene of interest replicates highly efficiently within the plant cell nucleus [268,269]. The geminivirus system considerably shortens the product recovery time and generates high yields of the desired antigen or antibody [270,271]. Further, a single given vector based on the BeYDV system can express several multimeric proteins in a non-competing manner whereas RNA virus-based systems require the use of multiple viruses that are non-competing to generate different proteins [269]. BeYDV vectors have a broad host range which enables protein expression at high yields in several dicot species [272].

A vital aspect of the prevention of rabies is therapy and the generation of robust rabies-neutralizing antibodies in plants is of great importance considering that the currently available anti-rabies sera produced in equines are limited in supply and of varied quality. van Dolleweerd et al., 2014 [273] demonstrated transgenic *N. tabacum* plants expressing the broadly neutralizing murine Mab E559 in its humanized IgG form as well as the murine version. These two antibodies assembled correctly and were found to be equivalent in activity to the MAbs produced in hybridomas, towards neutralizing the rabies virus. Moreover, the plant-expressed humanized antibody was found to be more efficient than the commercially available human rabies Ig vaccine (HRIG; Rabigam).

For the SARS-CoV-2, in contrast with the VLPs and virus subunit antigens which are designed to stimulate an immune response against the virus, recombinant antibodies could enable the slow-down of viral infection and thus provide the body sufficient time to elicit its own antibodies even before the infected individual succumbs to the Covid-19 disease. In this context, it is important to note that convalescent patient sera have been proved to diminish disease symptoms severity and promote recovery [175,274]. Thus, plants can be used as biofactories to express these antibodies that would act as reagents for both the detection of viruses and for enabling passive immunotherapy.

Three human-mouse chimaeric Mabs specific to the Ebola virus were generated in *N. benthamiana* plants through agroinfiltration using TMV-derived MagniCON viral vectors [262]. A cocktail of all the three MAbs (designated MB-003) which when administered to macaques at a dosage of 16.7 mg/kg per Mab 1 h after Ebola virus infection followed by booster doses at 4 and 8 days post-infection afforded 100% protection from challenge with lethal doses (1000 pfu) of the Ebola virus. Interestingly, these MAbs produced in plants were threefold more potent than those produced in CHO cells. Following the successful expression of the above MB-003 Mabs, Qiu et al., 2014 [264] documented the use of ZMapp, another mixture of anti-Ebola MAbs that included features of the ZMAb cocktail generated by the National Microbiology Laboratory of the Public Health Agency of Canada as well as another antibody medley. This ZMapp was developed by the San Diego Mapp Biopharmaceuticals along with Defyrus of Toronto and was synthesized by the Kentucky Bio-Processing. When tested in macaques, it was seen to afford 100% rescue from infection due to the Ebola virus challenge. It also proved to be highly therapeutic in reversing advanced stages of disease in many of the tested animals and led to complete recovery. Additionally, the ZMapp cocktail is recognized and bound to virions of the highly prevalent Guinean variant of the Ebola virus. The ZMapp far outperforms the effectiveness of any other anti-Ebola therapeutic described hitherto and this holds great promise towards its use for clinical prophylactic and therapeutic purposes. The efficacy of ZMapp was also tested in humans [275] and was found to be effective at treating the Ebola virus disease (Table 1).

Plants such as maize [276,277], tobacco [145], and rice [278] have been used as transgenic systems for expressing anti-HIV neutralizing antibodies 2G12 and 2F5 on a large scale. Fraunhofer IME has obtained a license for generating HIV 2G12 antibodies in tobacco for testing in Phase I human clinical trials and a similar strategy could be employed for producing antibodies capable of neutralizing SARS-CoV-2. Recently, rice has been used to express 2G12 as well as two antiviral lectins that would enable low-cost production of preformulated cocktails [152]. Singh et al., 2020 [279] report the expression of CAP256-VRC26 bNAbs (broadly neutralizing antibodies) against HIV with posttranslational modifications in *N. benthamiana* plants using MagnICON vectors. Co-expression of the tyrosyl protein sulfotransferase in these plants generated O-sulfated tyrosine in the heavy chain complementarity determining region (CDR) H3 loop of the bNAbs. These bNAbs showed structural folding similar to their mammalian cell-generated bNAb counterparts and exhibited equivalent neutralizing activity to the antibodies raised in mammalian cells. Moreover, these bNAbs showed high levels of potency against some of the subtype C HIV strains. This reveals the great potential of plant-derived systems for multiple post-translational engineering and produced fully active and viable bNAbs for use in passive immunization or as an alternative therapy for existing HIV/AIDS antiretroviral treatment regimens.

Additionally, therapeutic antibodies can be generated in plants in large amounts that can suppress the cytokine storm following infection by SARS-CoV-2 in several severe and fatal cases. Of these, tocilizumab/Actemra and sarilumab/Kevzara antibodies capable of binding to the interleukin-6 receptor (IL-6R) and promoted for therapy of rheumatoid arthritis can be repurposed for Covid-19 therapy. Currently, these therapeutic antibodies are in Covid-19 clinical trials [Long Island Press (2020), Swiss Broadcasting Corporation (2020)]. Interestingly, antibodies against the SARS-CoV-1 cross-react with SARS-CoV-2 and therefore, biologicals and monoclonal antibodies already generated against SARS-CoV-1 could be used to combat Covid-19 infection [280]. Therefore, besides producing prophylactic VLP-based vaccines, monoclonal antibodies produced in plants can provide a viable alternative to transfusion with convalescent plasma for safer intravenous administration in critically ill cases. Table 2 shows a list of plant-derived pharmaceuticals generated against a host of human viruses.

Table 3 shows a list of immune reactions elicited by plant virus-derived protein nanoparticle (VNP) vaccines.

## 11. Caveats of Plant-Derived Vaccines

When using plant-based edible vaccines, it is important to consider the following factors [13]: the plant material must have a long shelf life, be deliverable as raw material and be heat stable. The most popular grain crops for generating plant-based vaccines are rice and maize, while banana and tomato are most suitable as vegetative crops for the expression of vaccine candidates. The concerns of the public towards the use of GM crops particularly for edible plant vaccines also have to be factored in. Besides, GM crops should undergo strict human and environmental risk assessments to ensure their safety for widespread use. For parenterally delivered plant vaccines, the most economical method would be a suspension culture within a closed system in accordance with the GMP regulations. Nevertheless, plant-based vaccines are subject to the same regulatory practices as those of traditional vaccines. Factoring in all the above regulations, it seems these plant-based edible vaccines may not really be economical when compared with conventional vaccines. Research on plant-made vaccines is also clouded by concerns over the acceptance of GM plants. The administration of edible plant-based vaccines has several inherent caveats such as dosage requirement standardization/amount of food consumed, fruit ripeness, and whether the plant or fruit in question can be consumed raw or only after cooking which would denature the antigenic recombinant protein and thereby reduce the vaccine’s immunogenicity. Moreover, the fruits and plants are affected by infestation due to microbes, which would impact vaccine stability [305].

## 12. Risks of Plant-Made Vaccines

The technology of plant-based vaccines poses several risks to the environment such as gene transfer and undesired exposure to foreign antigens or proteins used as selectable markers [306]. There are potential dangers to human health including allergenicity due to inherent post-translational modifications, oral tolerance, inconsistent dosage, unwanted exposure to personnel working on the engineered plants as well as inadvertent exposure to the engineered antigens or selectable marker proteins within the food chain. However, these risks are surmountable through proper regulatory measures during all stages of production and dissemination of a potential plant-based candidate vaccine. Therefore, much of the success of this technology rests on the appropriate supervision and risk management by those involved as well as through setting quality standards for the manufacture of plant-based vaccines as enforced by the regulatory agencies. Nevertheless, there has to be a balance between the production/delivery of these vaccines on the one hand alongside the contingency and severity of potential risks on the other hand in the light of the price we have to pay for not deploying this highly promising technology.

## 13. Conclusions and Future Prospects

Recent research has proved that it is possible to use plants to regularly synthesize complex VLPs and antibodies in their fully assembled state that can stimulate prophylactic immunity or cause therapeutic effects towards disease-amelioration. The successful generation of antigen vaccine candidates for the HBV, HPV, and HIV viruses as well as the production of Influenza virus HA-only VLPs in plants have demonstrated the application of sophisticated transient expression technologies on industrial scales. In 1989, transgenic plants were for the first time used to produce antibodies. These antibodies can be conveyed orally, topically, or parenterally [307,308]. Plant-based antibody expression systems are capable of generating antibodies with the desired glycoforms [309] and plants that are glycol-engineered provide a much greater extent of glycan homogeneity. Moreover, the administration of antibodies through edible plant material would enable passive immunization through the stomach mucosa [310]. The use of biopharmed viral vaccines in animal husbandry and as part of emergency response vaccines and therapeutics in humans looks promising for the near future. The development and clinical testing of human prophylactic anti-viral vaccines involve lengthy times and rigorous analysis when compared to those of therapeutic vaccines and this poses huge impediments to the use of new biopharmed vaccines. There is a great need to generate biopharmed rapid response vaccines to respond to sudden outbreaks of emerging viral diseases and potential threats of bioterrorism. Increasingly, the generation of plant-based “biobetters” opens novel pathways to facilitate biopharming which is safe, rapid, and can be easily scaled up to manufacture high-value biopharmaceuticals and biologics. In the light of the increasing development of plant-derived biopharmaceuticals, regulatory agencies must enhance their knowledge about this newly emerging technology and adapt accordingly. Whether developing countries will reap the benefits of efficient, low-cost plant-based technologies in combating Covid-19 disease will be evident in the coming months.

## Figures and Tables

**Figure 1 vaccines-09-00761-f001:**
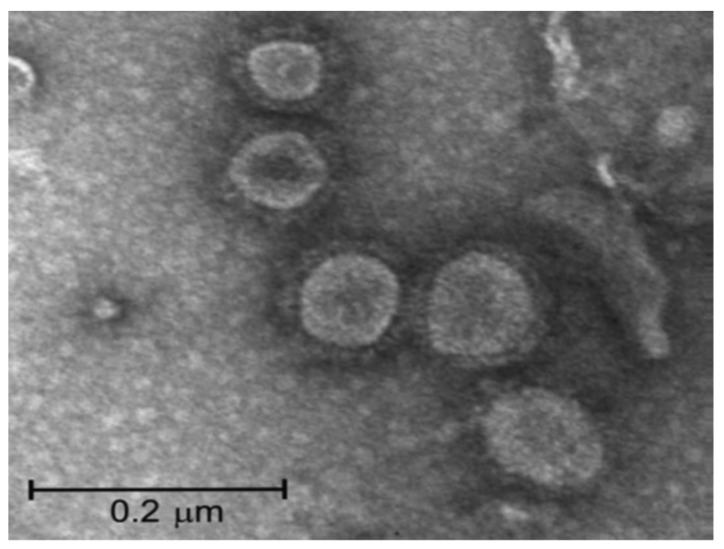
Image of H6-type influenza VLPs detected by transmission electron microscopy using negative staining. (Adapted from Smith et al., 2020 [116]).

**Figure 2 vaccines-09-00761-f002:**
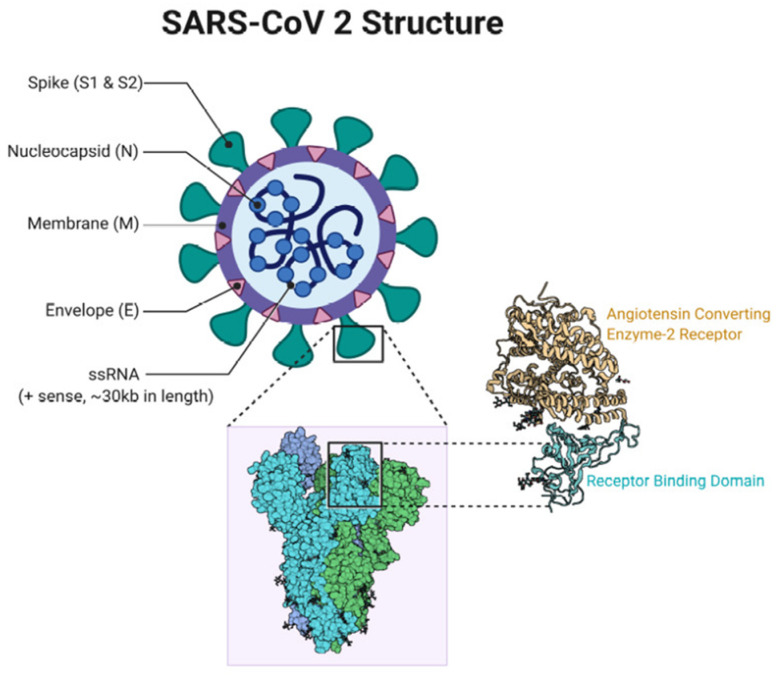
SARS-CoV-2 structure and its affinity with human ACE2 receptor (adapted from Mahmood et al., 2021 [184]).

**Figure 3 vaccines-09-00761-f003:**
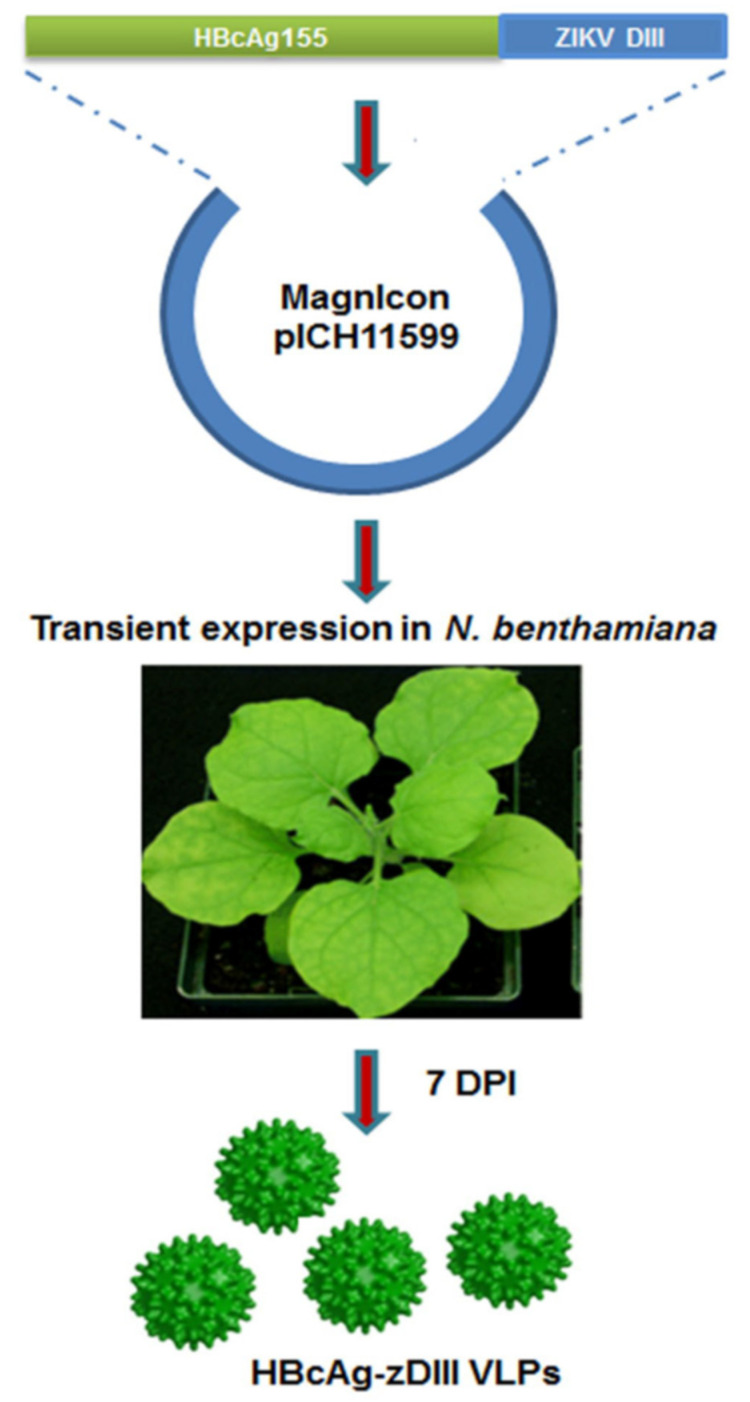
Use of the MagnICON vector for expressing HBcAg-zDIII protein in *N. benthamiana*. The zDIII coding sequence was genetically fused to the 3′ end of the HBcAg gene (amino acid 1–155) and then cloned into the MagnICON-based plant expression vector pICH11599. This construct was transformed into *Agrobacterium tumefaciens* which was infiltrated into leaves of *N. benthamiana* to enable transient expression. At 7 days post-inoculation, the leaves were harvested to isolate the HBcAg-zDIII fusion protein. (Adapted from Yang et al., 2017 [14]).

**Table 1 vaccines-09-00761-t001:** Plant-based viral vaccines currently under clinical trials.

Viral Vaccine	Antigen/Expression System	Stage of Clinical Trial	Reference
Quadrivalent Influenza vaccine	Mix of recombinant H1, H3, and two B hemagglutinin proteins expressed as VLPs transiently in *N. benthamiana*	Phase 3 ongoing	[81,82]
SARS-CoV-2 vaccine	Recombinant spike (S) glycoprotein expressed as VLPs transiently in *N. benthamiana*	Phase 1 successfully completed; Phase 2/3 ongoing	[83]
Ebola virus vaccine	ZMapp produced by rapid transient expression of 3 plant-based neutralizing monoclonal antibody cocktails against Ebola; made in *N. benthamiana*	Phase 2/3 completed; FDA approved	[84]
Rotavirus vaccine	Four structural antigens of rotavirus (VP2, VP4, VP6, and VP7) expressed as VLPs in plants	Phase 1	[85]
Norwalk virus vaccine	CP expressed in potato	Early phase 1	[86]
Rabies virus vaccine	GP/NP antigens expressed in spinach	Early phase 1	[87]
Hepatitis B virus vaccine	HBsAg expressed in lettuce	Early phase 1	[88]
Hepatitis B virus vaccine	HBsAg expressed in potato	Phase 1	[89]
Bacterial vaccine: *Vibrio cholera* vaccine	CTB antigen expressed in rice	Phase 1	[90,91]

(Adapted from Kurup and Thomas, 2020 [92]).

**Table 2 vaccines-09-00761-t002:** Some examples of plant-based biopharmaceuticals against human viruses.

Virus.	Plant-Derived Biopharmaceutical	Technology for Expression In Plants	Reference
Influenza	VLPs	Plant virus vector	Yusibov et al., 2015 [56] Marsian and Lomonosoff, 2016 [281] D’Aoust et al., 2010 [282] Lindsay et al., 2018 [95] Marquez-Escobar et al., 2017 [4]
Ebola	mAbs	Transgenic plants, plant virus vector	McCarthy, 2014 [283] Zeitlin et al., 2011 [258] Monreal-Escalante et al., 2017 [284] Rosales-Mendoza et al., 2017 [285] Phoolcharoen et al., 2011 [286]
HIV	Griffithsin	Transgenic plants	Hoelscher et al., 2018 [287] Vamvaka et al., 2018 [152]
HIV	VLPs	Transiently and transgenic plants	Cervera et al., 2019 [288]
HIV	Antigens derived from Env, Gag proteins	Transiently and transgenic plant	Rosales-Mendoza et al., 2012 [289]
HIV	mAbs	Transgenic plant	Lotter-Stark et al., 2012 [290]
West Nile, Zika, Chickungunya	mAbs	Transgenic plants	Rybicki, 2017 [291]
WNV, Zika	Envelope protein	Transgenic plant	Yang et al., 2018 [292] Lai et al., 2018 [293]

(Adapted from Hefferon, 2019 [294]).

**Table 3 vaccines-09-00761-t003:** Immune reactions induced by plant virus-derived protein nanoparticle (VNP) vaccines.

Scaffold Platform	Viral Disease	In Vitro/In Vivo	Elicited Immune Response	Specific Immune Reactions	References
CPMV	HIV-1	In vivo	Humoral	Neutralizing antibodies	McInerney et al. (1999) [295] Durrani et al. (1998) [296]
4	HIV-1	In vivo	Cellular	Antigen-specific T-cell proliferation	McInerney et al. (1999) [295]
PVX	HIV-1	In vivo	Humoral	Neutralizing antibodies	Marusic et al. (2001) [297]
	Influenza A {H1N1)	In vivo	Cellular	Antigen-specific CD8+ T-cell activation	Lico et al. (2009) [298]
	HCV	In vitro, In vivo	Humoral	Antigen-specific Abs	Uhde-Holzem et al. (2010) [299]
TMV	Influenza (H1N1)	In vivo	Humoral	Antigen-specific Abs, protection against challenge	Mallajosyula et al. (2014) [93]
CMV	HCV	In vitro, In vivo	Humoral	Antigen-specific Abs	Piazzolla et al. (2005) [53] Nuzzaci et al. (2007) [54]
	HCV	In vitro, In vivo	Cellular	Cytokine release (IFN-γ, IL-12, and IL-15)	Piazzolla et al. (2005) [53] Nuzzaci et al. (2007) [54]
	Zika virus	In vitro, In vivo	Humoral	Neutralizing antibodies	Cabral-Miranda et al. (2017) [300]
AlMV	RSV	In vivo	Humoral	Antigen-specific Abs	Yusibov et al. (2005) [301]
	RSV	In vivo	Cellular	CD4+ and CD8+ T-cell responses	Yusibov et al. (2005) [301]
TBSV	HIV-1	In vitro, In vivo	Humoral	Antigen-specific Abs	Joelson et al. (1997) [302]
PapMV	Influenza	In vitro, In vivo	Humoral	Antigen-specific Abs, B-cell expansion, protection against challenge	Denis et al. (2008) [109] Hanafi et al. (2010) [303] Bolduc et al. (2018) [108] Carignan et al. (2015) [106] Therien et al. (2017) [107]
	Influenza	In vitro, In vivo	Cellular	Antigen-specific CD8+ T-cell expansion and response	Babinet al. (2013), [110] Leclercet al. (2007) [112] Hanafi et al. (2010) [303] Laliberte-Gagne et al. (2019) [111]
	HCV	In vivo	Humoral	Antigen-specific Abs	Denis et al. (2007) [48]

(Adapted from Butkovich et al., 2021 [304]).

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
