# Peer review of "Combating Human Viral Diseases: Will Plant-Based Vaccines Be the Answer?"

_vaccines, 2021, doi:10.3390/vaccines9070761_

Round 1
Reviewer 1 Report
Venkataraman et al. discussed the potential of plant-based vaccines, which was based on the technology of application of plants and plant cell culture to manufacture high-value recombinant proteins. In each section, the authors thoroughly discussed the possible implementation of plant-based vaccines on both DNA and RNA viruses such as hepatitis B virus (HBV), hepatitis C virus (HCV), the cancer-causing virus human papilloma virus (HPV), human immunodeficiency virus (HIV), influenza virus, zika virus, and severe acute respiratory syndrome coronavirus 2 (SARS-CoV-2). The authors also described the caveats of plant-based molecular pharming and future perspectives of this technology.
This review contains enormous work on collecting the relevant references, each section includes very detailed introduction, and most importantly the latest vaccine developments in the generation of plant-based viral vaccines. Overall, it is a nice review article, I just have a few concerns and comments as listed below.
- The plant-based vaccines sound quite attractive, based on its fast production of engineered proteins at low cost. Although the authors described the caveat of plant-based vaccines, did not mention the potential risks of it. For example, the plant-based vaccines might induce allergenicity, since post-translational modifications might induce allergic responses. The plant-based vaccines, especially recombinant genes might influence the environment. Therefore, the authors might need to mention the potential risks of plant-based vaccines.
- HIV vaccines: the genome organization of HIV-1 genome is not accurate, please correct Line 446 to 449 as following. The HIV genome contains 9 genes that encode fifteen viral proteins. 9.2 kb unspliced genomic transcript encodes for Gag, Pol, and Env (just for authors: Pol codes for reverse transcriptase (RT), RNase H, integrase (IN), and HIV protease, therefore it should not be Gag, RT, and Pol). The authors emphasized the efficient production of HIV-1 envelope proteins, however, the scientific challenges for the HIV-1 vaccine development are enormous. As the authors mentioned, first the high mutation rates of HIV-1 envelope protein (viral diversity), thus the vaccine protection will necessarily be dependent on the capacity of immune responses to cross-react with highly heterologous viruses. Many HIV-1 vaccines could not induce broadly reactive neutralization antibody responses as well as by their inability to elicit CD8+ T lymphocyte responses.
Taken together, I would suggest the authors delete this HIV-1 vaccine section or re-write this part.
- Please add the full name for SARS-CoV-2, which is severe acute respiratory syndrome coronavirus 2.
Author Response
Reviewer #1:
Venkataraman et al. discussed the potential of plant-based vaccines, which was based on the technology of application of plants and plant cell culture to manufacture high-value recombinant proteins. In each section, the authors thoroughly discussed the possible implementation of plant-based vaccines on both DNA and RNA viruses such as hepatitis B virus (HBV), hepatitis C virus (HCV), the cancer-causing virus human papilloma virus (HPV), human immunodeficiency virus (HIV), influenza virus, zika virus, and severe acute respiratory syndrome coronavirus 2 (SARS-CoV-2). The authors also described the caveats of plant-based molecular pharming and future perspectives of this technology.
This review contains enormous work on collecting the relevant references, each section includes very detailed introduction, and most importantly the latest vaccine developments in the generation of plant-based viral vaccines. Overall, it is a nice review article, I just have a few concerns and comments as listed below.
1. The plant-based vaccines sound quite attractive, based on its fast production of engineered proteins at low cost. Although the authors described the caveat of plant-based vaccines, did not mention the potential risks of it. For example, the plant-based vaccines might induce allergenicity, since post-translational modifications might induce allergic responses. The plant-based vaccines, especially recombinant genes might influence the environment. Therefore, the authors might need to mention the potential risks of plant-based vaccines. There are several risks during the production and delivery stages of this technology, with potential impact on the environment and on human health.
A paragraph on the risks of plant-made vaccines is included.
2. HIV vaccines: the genome organization of HIV-1 genome is not accurate, please correct Line 446 to 449 as following. The HIV genome contains 9 genes that encode fifteen viral proteins. 9.2 kb unspliced genomic transcript encodes for Gag, Pol, and Env (just for authors: Pol codes for reverse transcriptase (RT), RNase H, integrase (IN), and HIV protease, therefore it should not be Gag, RT, and Pol). The authors emphasized the efficient production of HIV-1 envelope proteins, however, the scientific challenges for the HIV-1 vaccine development are enormous. As the authors mentioned, first the high mutation rates of HIV-1 envelope protein (viral diversity), thus the vaccine protection will necessarily be dependent on the capacity of immune responses to cross-react with highly heterologous viruses. Many HIV-1 vaccines could not induce broadly reactive neutralization antibody responses as well as by their inability to elicit CD8+ T lymphocyte responses.
Taken together, I would suggest the authors delete this HIV-1 vaccine section or re-write this part.
Lines 446 to 449 have been deleted.
3. Please add the full name for SARS-CoV-2, which is severe acute respiratory syndrome coronavirus 2.
The full name of SARS-CoV-2 has been mentioned at the first instance where it is addressed in the manuscript.

Reviewer 2 Report
The manuscript “Combating human viral diseases: Will plant-based vaccines be the answer?” by Venkataraman and colleagues is a comprehensive review article about the production of vaccines by plants. The manuscript is very well written, well organized, and well presented. This is an exceptional review article. I have no concerns with this manuscript.
Minor comments:
1) Throughout the text, the authors mention several vaccines produced by plants, currently undergoing clinical trials. Could the authors please include a table referring to the vaccines produced in plants that are currently in clinical trials?
2) Given the global relevance tuberculosis and malaria, could the authors please include a short paragraph in the discussion, about the possibility of vaccines against these - and other diseases not mentioned in the manuscript - being produced by plants?
3) Line 75, please replace “(“ with “[“.
4) Lactuca sativa shows in the text for the first time in line 151, not 214. Please transfer the “(lettuce)” to line 151.
5) In lines 139, 160, 167, 324, 334, 338, 347, and 398, Nicotiana benthamiana should be in italic. Please verify all species throughout the manuscript and make sure these are in italic.
6) Also, the first time a species name appears in the manuscript, should be without abbreviation. Therefore, in line 139, should be Nicotiana benthamiana, and subsequently, N. benthamiana.
7) Line 619-620: please remove the hyperlink, and add a reference number.
8) Lines 649-651: please remove this statement, as this is unpubished data.
9) Lines 659-661: please remove this statement, as this is unpubished data.
Author Response
Reviewer #2:
The manuscript “Combating human viral diseases: Will plant-based vaccines be the answer?” by Venkataraman and colleagues is a comprehensive review article about the production of vaccines by plants. The manuscript is very well written, well organized, and well presented. This is an exceptional review article. I have no concerns with this manuscript.
Minor comments:
1) Throughout the text, the authors mention several vaccines produced by plants, currently undergoing clinical trials. Could the authors please include a table referring to the vaccines produced in plants that are currently in clinical trials?
A new table (Table 3) on plant-based vaccines currently under clinical trials has been included.
2) Given the global relevance tuberculosis and malaria, could the authors please include a short paragraph in the discussion, about the possibility of vaccines against these - and other diseases not mentioned in the manuscript - being produced by plants?
A separate section on plant-based vaccines against malaria, tuberculosis and dengue has been included under the heading “Other plant-made vaccines”.
3) Line 75, please replace “(“ with “[“.
“(“ has been replaced with “[“
4) Lactuca sativa shows in the text for the first time in line 151, not 214. Please transfer the “(lettuce)” to line 151.
The word lettuce has been moved from line 214 to line 151.
5) In lines 139, 160, 167, 324, 334, 338, 347, and 398, Nicotiana benthamiana should be in italic. Please verify all species throughout the manuscript and make sure these are in italic.
All species names have been changed to italic.
6) Also, the first time a species name appears in the manuscript, should be without abbreviation. Therefore, in line 139, should be Nicotiana benthamiana, and subsequently, N. benthamiana.
All mentions of genus names have been elaborated in the first instance followed by abbreviation in the subsequent instances.
7) Line 619-620: please remove the hyperlink, and add a reference number.
Line 619-620: hyperlink removed and reference number added.
8) Lines 649-651: please remove this statement, as this is unpubished data.
Lines 649-651 have been removed.
9) Lines 659-661: please remove this statement, as this is unpubished data.
Lines 659-661 have been removed.
